# TopicGAN: Unsupervised Text Generation from Explainable Latent Topics

## Abstract

Learning discrete representations of data and then generating data from the discovered representations have been increasingly studied because the obtained discrete representations can benefit unsupervised learning. However, the performance of learning discrete representations of textual data with deep generative models has not been widely explored. In addition, although generative adversarial networks(GAN) have shown impressing results in many areas such as image generation, for text generation, it is notorious for extremely difficult to train. In this work, we propose TopicGAN, a two-step text generative model, which is able to solve those two important problems simultaneously. In the first step, it discovers the latent topics and produced bag-of-words according to the latent topics. In the second step, it generates text from the produced bag-of-words. In our experiments, we show our model can discover meaningful discrete latent topics of texts in an unsupervised fashion and generate high quality natural language from the discovered latent topics.

## 1 Introduction

Recently, deep generative models (Goodfellow et al., 2014; Kingma & Welling, 2013; Makhzani et al., 2015) have achieved a great success on generating realistic images, videos and audio. Learning discrete representations of data and then generating data from the discovered representations have been increasingly studied, because the obtained discrete representations can benefit unsupervised learning (Chen et al., 2016; van den Oord et al., 2017), semi-supervised learning (Odena et al., 2016), and few-shot learning. However, it remains extremely challenging for generating texts and learning interpretable discrete representations of texts due to the discrete, sparse and high dimensional properties of textual data.

Arjovsky et al. (2017) mentioned that the original generative adversarial network (GAN) fails to generate discrete data due to the gradient vanishing problem. Wasserstein distance has been proposed to more precisely measure the distance between real and fake distributions and therefore tackles the gradient vanishing problem (Arjovsky et al., 2017; Gulrajani et al., 2017). Another obstacle for generating discrete data is the non-differentiable function when generating words, which makes the gradients unable to be backpropagated from the discriminator to the generator. With the help of reinforcement learning, the generator is able to maximize the scores from the discriminator when the gradient is not able to flow from discriminator (Yu et al., 2017; Li et al., 2017; Fedus et al., 2018).

In natural language processing (NLP), learning representations of texts is shown useful for unsupervised language understanding. While learning continuous text representations has been widely studied (Kiros et al., 2015; Arora et al., 2017; Logeswaran & Lee, 2018), learning high-level discrete representations has been explored by fewer work (Zhao et al., 2018; Miao et al., 2016). In order to take non-differentiable discrete variables as latent representations of an auto-encoder, some special methods such as Gumbel-Softmax (Jang et al., 2016) or vector quantisation (van den Oord et al., 2017) were applied by the prior work. Furthermore, because textual data is high dimensional and has rich but sometimes noisy information such as stop words, it is challenging to learn useful discrete representations of texts.

To mitigate the difficulty of text generation and text discrete representation learning, this paper proposes TopicGAN, which simplifies those two problems by a two-step progressive generation. The idea of dividing the generation process into a pipeline yielded impressive results on high resolution

image generation (Zhang et al., 2016; Karras et al., 2018). The progressive generation is a natural way to generate texts considering how human writes texts. When writing articles, human first considers the context of the texts and then organizes the context with correct grammar. Hence, in this work, we split text generation into two step, one is context generation and another is text generation from context with correct grammar. In the first step, we use topic model to discover the latent topics and use bag-of-words generator to produce bag-of-words according to the discovered latent topics and continuous noise. In the second step, based on the generated bag-of-words, we decode a sequential text by a recurrent neural network (RNN).

We utilize InfoGAN (Chen et al., 2016) to discover the latent topics and generate a bag of topical words without supervision in the first step, where the categorical classifier of InfoGAN can be considered as a topic model, which is able to discover latent topics of documents. We show that our model can yield explainable topics and outperform previous topic models such as latent Dirichlet allocation (LDA) (Blei et al., 2003) or variational topic model (Miao et al., 2016) for unsupervised document classification. Topic modeling can be applied to many applications, including extractive text summarization(Titov & McDonald, 2008), document retrieval (Wei & Croft, 2006) or unsupervised classification. Also, unlike previous topic models that did not consider the word correlation during generating words, our method is able to regularize the correlation between words by the discriminator so that it yields more reasonable bag-of-words and produces high-quality texts.

The contributions of this work are three-fold:

- We propose two-step progressive text generation, which aligns well with the nature of text generation.

- Our model is able to discover explainable topics by a topic classifier and achieves promising performance on unsupervised learning.

- Compared to previous topic models, the proposed TopicGAN is able to capture the correlation between words, and thus performs better generation results.

## 2 RELATED WORK

**Text Generation via GAN**   Prior work attempted at generating texts using GAN (Yu et al., 2017; Che et al., 2017; Li et al., 2017; Liu et al., 2018), and there are two main directions. One is to tackle the gradient vanishing problem in the original GAN, where JensenShannon divergence to evaluate the discrete real data distribution and the continuous generated distribution. By using Wasserstein distance, the discrete and continuous distributions can be properly measured. However, the Wasserstein distance requires the discriminator to be a Lipschitz continuous function; therefore some restrictions including weight-clipping (Arjovsky et al., 2017), gradient penalty (Gulrajani et al., 2017) are imposed on the discriminator.

Another direction is to feed sampled discrete words from generated distribution to the discriminator. While the sampling operation is non-differentiable, reinforcement learning is applied to optimize the score from the discriminator. Some designed reward functions, such as Monte Carlo tree search (MCTS) (Yu et al., 2017), are proposed to evaluate the generated word for each time step (Lin et al., 2017; Fedus et al., 2018; Li et al., 2017). Our proposed progressive two-step generation framework can be easily combined with any current adversarial text generation models, because we can choose one those current work as our final jointly optimization method. Our framework effectively facilitates the training process of text generation by decomposing the task into two subproblems.

**InfoGAN**   InfoGAN has shown impressive performance for learning disentangled representations of images in an unsupervised manner (Chen et al., 2016). The original GAN generates images from a continuous noise $z$, while each dimension of the noise does not contain disentangled features of generated images. To learn semantically meaningful representations, InfoGAN maximizes the mutual information between input code $c$ and the generated output $G(z, c)$. However, maximizing the mutual information is intractable, because it requires the access of $P(c \mid G(z, c))$. Based on variational information maximization, Chen et al. (2016) used an auxiliary distribution $Q(c \mid G(z, c))$ to approximate $P(c \mid G(z, c))$. The auxiliary function $Q$ can be a neural network that can be jointly

Figure 1: The architecture illustration of the proposed model. In the upper part (Bag-of-Words Generator), the $G_{bow}$ takes discrete one-hot topic code $c$ and continuous noise vector $z$ as input, and generates bag-of-words. The $D_{bow}$ discriminates its input bag-of-words is from $G_{bow}$ or from human-written text. The topic model $Q$ predicts the latent topic of input bag-of-words. The noise predictor predicts the noise of input bag-of-words. In the lower part (Word Sequence Generator), the sequence generator $G_{seq}$ takes bag-of-words from $G_{bow}$ as input and generates text. After training $G_{bow}$ and $G_{seq}$ separately, we train upper part and lower part jointly. During joint training, we add an extra sequential text discriminator $D_{seq}$ which takes text and bag-of-words as input and discriminates the input bag-of-words and text pair is real or fake.

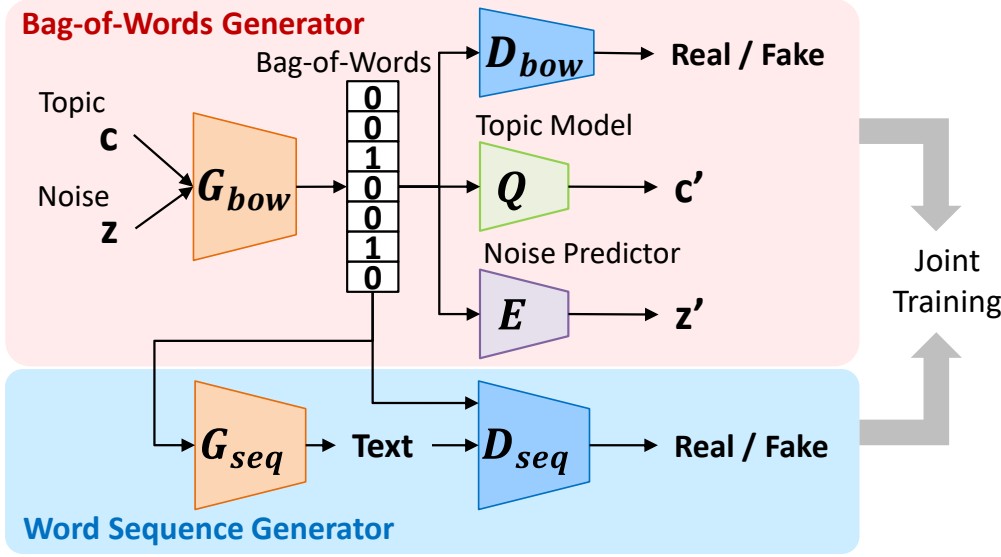

optimized:

$$\min_{G,Q} \max_{D} \mathbb{E}_{x \sim P_{data}}[\log(D(x))] + \mathbb{E}_{z \sim P_z, c \sim P_c}[\log(1 - D(G(z,c))) - \lambda Q(c \mid G(z,c))], \quad (1)$$

where $P_{data}$ is the real data distribution, $P_z$ is the noise distribution, and $P_c$ is the code distribution. The code $c$ can be either continuous or categorical. In our work, the code $c$ is set to be categorical, and the categorical classifier $Q$ becomes a topic model. Therefore, we call the third term in (1) categorical loss.

## 3 TOPICGAN

When using GAN for natural language generation, the generator has difficulty generating text with reasonable context and correct grammar simultaneously. In addition, when feeding a sequential text to the InfoGAN categorical code classifier, due to its complex structure, the classifier fails to discover meaningful discrete information. The key idea of this work is that we divide text generation into two steps: 1) generating bag-of-words that can roughly represent the topical information, and then 2) generating sequential texts from the learned topical words. As shown in Figure 1, given a discrete topic $c$ and a continuous noise $z$ as the input, the bag-of-words generator $G_{bow}$ generates topical words. After obtaining topical words, the sequence generator $G_{seq}$ generates texts according to topical words.

### 3.1 GENERATING TOPICAL WORDS

The upper part of Figure 1 illustrates how our model generate topical words, where there are a bag-of-words generator $G_{bow}$, a bag-of-words discriminator $D_{bow}$, a topic model $Q$, and a noise predictor $E$.

- Bag-of-words generator $G_{bow}$
  It takes a discrete one-hot topic code $c$ and a continuous noise $z$ as the input, and manages to generate bag-of-words that captures the input topical information and are indistinguishable from the bag-of-words of real texts. Here the bag-of-words is a binary vector generating from $sigmoid$ function, where each dimension indicates whether a single word in the dictionary exists in the text.

- Bag-of-words discriminator $D_{bow}$
  It takes the bag-of-words vector as its input and distinguishes whether it is generated or human-written.

- Topic model $Q$
  It is a categorical topic code classifier that is implemented by a matrix, considering that a linear model is easier to interpret the generated bag-of-words.

- Noise predictor $E$
  It focuses on predicting the noise that can reconstruct the input bag-of-words by $G_{bow}$.

Similar to Chen et al. (2016), we apply (1) to train our generator $G_{bow}$, discriminator $D_{bow}$, and the topic model $Q$. However, it is difficult to generate discrete bag-of-words due to the gradient vanishing problem, we apply WGAN (Arjovsky et al., 2017) loss to train $G_{bow}$ and $D_{bow}$.

In addition, during training, there is a severe *mode collapse* issue within the same topic. That is, given the same topic code, the generator ignores the continuous noise and output the same topical words. The reason is that outputting the same bag of words for the generator is the optimal solution to maximize the mutual information between the discrete input topic code and its output. To tackle this issue, we clip the probability from topic model $Q$ in (1) to a specific range $\alpha$, and rewrite the (1) to:

$$\min_{G,Q} \max_{D} \mathbb{E}_{x \sim P_{data}}[D_{bow}(x)] - \mathbb{E}_{z \sim P_z, c \sim P_c}[D_{bow}(G_{bow}(z,c))] -$$
$$\lambda \mathbb{E}_{z \sim P_z, c \sim P_c}[\min(Q(c \mid G_{bow}(z,c)), \alpha)]. \tag{2}$$

, where $x$ is the binary bag-of-words vector of real text. Here, to constrain the $D_{bow}$ to be Lipschitz function, we apply gradient penalty (Gulrajani et al., 2017) to $D_{bow}$. In addition, we apply batch-normalization to alleviate the mode collapse problem (Ioffe & Szegedy, 2015).

In order to obtain better results, an auto-encoder is included in the optimization procedure (Larsen et al., 2015; Huang et al., 2018). Here we use binary cross entropy loss as reconstruction loss function:

$$\min_{G,Q,E} \mathbb{E}_{x \sim P_{data}}[-x * \log(G_{bow}(Q(x), E(x))) + (1-x) * \log(1 - G_{bow}(Q(x), E(x)))] \tag{3}$$

, where the topic classifier $Q$ and the noise predicor $E$ encode real text bag-of-words $x$ into the discrete code and continuous noise respectively. We train 2 and 3 alternately.

## 3.2 GENERATING TEXTS FROM TOPICAL WORDS

The lower part of Figure 1 illustrates how the model generates natural language, where there are a sequence generator $G_{seq}$ and a sequence discriminator $D_{seq}$.

- Sequence generator $G_{seq}$
  After obtaining bag of topical words, we use an LSTM model to generate sequential texts from the bag-of-words. The bag-of-words vector is fed into a feedforward neural network and the output of feedforward neural network is used to initialize the hidden state of LSTM. We use an extra vector as the LSTM input to keep track of which input bag-of-words have been generated in order to avoiding generating the same words repeatedly.

- Sequence discriminator $D_{seq}$
  We introduce a sequence discriminator $D_{seq}$ that encourages $G_{seq}$ to produce realistic sequential texts. If $D_{seq}$ simply takes sequential text as the input, it may make the sequence generator generate texts that is realistic but unrelated to the input bag-of-words of Therefore, conditioned on the generated bag-of-words, $D_{seq}$ is also able to discriminate whether the text is produced from input bag-of-words. The pairs of bag-of-words of human written text and corresponding text are regarded as real data for discriminator.

Table 1: Unsupervised classification accuracy

| Methods | 20NewsGroups | Yahoo! Answers | DBpedia |
|---------|--------------|----------------|---------|
| LDA | 29.78 | 25.95 | 68.42 |
| NVDM | 23.98 | — | — |
| TopicGAN | **41.01** | **42.14** | **83.73** |

**Supervised pretraining of sequence generator** As each sequential text has its corresponding bag-of-words, we can obtain numerous (bag-of-words, text) pairs to pretrain $G_{seq}$. To make $G_{seq}$ robust to the noisy bag-of-words input, during pretraing, we add some noise such as randomly deleting words to the input texts.

**Joint Training.** After pretraining $G_{seq}$, we jointly optimize the whole model by adversarial training between $G_{seq}$ and $D_{seq}$. In the joint training, two steps are jointly trained that we train all the modules to directly generate text conditioned on topic $c$ and noise $z$. We can choose any existing adversarial sequence generation method such as Yu et al. (2017) or Gulrajani et al. (2017) to jointly train $G_{seq}$ and $D_{seq}$. In our work, the $D_{seq}$ is a deep residual network which takes the output of $G_{seq}$ as input. We simply apply the training procedure of WGAN-gp (Gulrajani et al., 2017) to train $G_{seq}$ and $D_{seq}$.

## 4 EXPERIMENTS

In this section, we evaluate whether our method is able to learn meaningful latent topics, and show that our two-step progressive generation can generate high-quality texts. For all experiments, in the first step, we set our bag-of-words vocabulary size to 3k and removed stopwords. By setting smaller vocabulary size in the first step, the topic classifier discovered better topics. When generating texts in the second step, we set the vocabulary size to 15k.

### 4.1 TOPIC MODELING RESULTS

To evaluate the quality of the learned latent topics, we test whether the topic classifier $Q$ is able to correctly discover the latent class same as the class labeled by human. We evaluate the topic classifier on three datasets including 20NewsGroups, DBpedia ontology classification dataset and Yahoo! answers.

The 20NewsGroups is a news classification dataset composed of 20 different classes of news with 11,314 training and 7,531 testing documents. DBpedia ontology classification dataset is constructed by Zhang & LeCun (2015). They selected 14 ontology classes from DBpedia 2014, and for each class they randomly picked 40,000 training samples and 5,000 testing samples. Thus, there are total 560,000 and 70,000 training and testing samples respectively. Yahoo! answers is a question type classification dataset with 10 types of question-answer pairs constructed by Zhang & LeCun (2015). There are 1,400,000 training samples and 60,000 testing samples.

#### 4.1.1 UNSUPERVISED CLASSIFICATION

For all experiments including baseline methods, we set the number of latent topics same as the number of true classes in each dataset. We used the topic classifier $Q$ to predict the latent topic distribution of each sample, and we assigned each sample to the latent topic with the maximum probability. The samples within a latent topic cluster used its true label to vote for which label should be assigned to the whole cluster. After assigning each latent topic to its corresponding label, we evaluate the classification accuracy as the quality of the captured latent topics.

We compared our method with a statistical topic model, LDA (Blei et al., 2003), and a variational topic model(NVDM) (Miao et al., 2016). The results of unsupervised classification are shown in Table 1, where compared to previous topic models, our model significantly outperforms baselines for unsupervised classification.

The main reason that our method achieves better result on unsupervised classification is that LDA asuumes each documents are produced from mixture of topics, while our method assumes that the documents are produced from a single topic and a noise controlling the difference of texts within a topic. In unsupervised document classification, the documents have only one single class. Therefore,

Table 2: Ablation study on unsupervised classification.

| Methods | 20NewsGroups | Yahoo! Answers | DBpedia |
|---|---|---|---|
| TopicGAN | 41.01 | 42.14 | 83.73 |
| TopicGAN w/ one hidden layer | 36.89 | 42.14 | 71.26 |
| TopicGAN w/o loss clipping | 35.83 | 33.46 | 74.12 |
| TopicGAN w/o auto-encoder | 23.11 | 20.66 | 62.43 |

Table 3: Topic coherence scores. Higher is better.

| Methods | 20NewsGroups | Yahoo! Answers | DBpedia | Gigaword |
|---|---|---|---|---|
| LDA | 42.45 | 34.79 | 50.33 | 33.54 |
| TopicGAN | 43.16 | 39.04 | 51.87 | 40.17 |

for those unsupervised classification experiments, assuming each documents coming from a single main topic is a more appropriate assumption, which allows our model to learn more distinct topics. In addition, as the length of our training documents are short, its hard to break the short text into several topics, which is one of the possible reason that makes LDA works not well on short text.

### 4.1.2 ABLATION STUDY

In this section we show that all mechanisms described in Section 3.1 including categorical loss clipping, training with auto-encoder are useful for unsupervised classification. As shown in Table 1, the key trick that greatly improved the performance is auto-encoder training of bag-of-words generator $G_{bow}$ and topic model $Q$. Without training with auto-encoder, there was almost only half of the original performance in 20NewsGroups and Yahoo! Answers datasets. Categorical loss clipping also improved the performance. It also alleviated the mode collapse problem within a single class and made the training process more stable.

The model complexity of topic classifier also influenced the classification accuracy. In the simpler dataset like 20 news or DBpedia, using a single matrix as model of discriminator (Table.1 no hidden layer) yielded better performance. While in more difficult dataset like Yahoo! Answers dataset, topic classifier with one hidden feed forward layer performed slightly better.

### 4.2 TOPIC COHERENCE

In Section 4.1, we find our method can outperform previous methods on unsupervised text classification. In this section, we discuss the quality of learned topic words on quantitative analysis and qualitative analysis. The topic words of our topic model can be retrieved as following. We used a single matrix $M_{V \times K}$ as our topic classification model $Q$, where V is the bag-of-words vocabulary size and K is the latent topic number. The value of $M_{v,k}$ represents the importance of v-th word to k-th topic. The top few words with higher values within each column are selected as its topical words.

**Quantitative analysis.** We use the $C_v$ metric (Roder et al., 2015) to evaluate the topic coherence score. The coherence score is computed by using English Wikipedia of 5.6 million articles as external corpus. Table 3 lists the coherence score on different datasets. Compared to LDA, TopicGAN is on par with LDA on DBpedia on, and outperforms LDA on all other datasets. This result suggests the effectiveness of using info-GAN and neural network to train a explainable topic model.

**Qualitative analysis.** To further analyze the quality of discovered topics, the top 20 topical words of latent topics are listed inTable 6 and the generated corresponding texts are shown in Table 7. From the tables, the captured latent topics are clearly semantically different based on the generated topical words. Similarly, the generated texts are also fluent and topically related to the associated topics.

### 4.3 TEXT GENERATION RESULTS

We conducted sequential text generation experiments on two datasets including DBpedia and English Gigaword. English Gigaword is a summarization dataset which is composed of first sentence of articles and their corresponding titles. The pre-process script (Rush et al., 2015) yielded 3.8M

Table 4: Perplexity of generated langauge on DBpedia and English Gigaword data. As there is no labeled class in English Gigaword, we cannot train class LM in English Gigaword.

| Method | DBpedia | | English Gigaword |
|---|---|---|---|
| | Class LM | General LM | General LM |
| Training set | 26.15 | 33.55 | 36.07 |
| Testing set | 32.69 | 35.53 | 43.50 |
| VAE+WGAN-gp | — | 33.67 | 50.55 |
| TopicGAN | 31.28 | 34.16 | 55.73 |
| TopicGAN (Joint Training) | 30.78 | 34.09 | 54.32 |

Table 5: (a) The accuracy of human correctly classified texts; (b) the ratio of human preference for the generated texts.

(a)

| | Accuracy |
|---|---|
| Topic 1 | 96.2 |
| Topic 2 | 98.6 |
| Average | 97.4 |

(b)

| Method | Preference |
|---|---|
| VAE+WGAN-gp | 35.01% |
| TopicGAN | 28.32% |
| TopicGAN (Joint Training) | 36.67% |

training samples and 400K validation samples. We trained our model to generate the first sentence of articles on training set. Unlike DBpedia which has the labeled classes, English Gigaword has no labeled classes. Therefore, we conducted human evaluation to evaluate whether our model is able to generate text from meaningful discovered topics.

We compare our method with the baseline method which was pre-trained with VAE and then fine tuned by WGAN-gp. Instead of generating text conditioned on discrete code and continuous noise, the baseline method was simply conditioned on a continuous noise. When pre-training variational auto-encoder, the tricks like KL-term annealing (Semeniuta et al., 2017) were applied. We chose this method as our baseline because our sequence generator $G_{seq}$ were pre-trained with bag-of-words to text language model, and we expected our baseline also pre-trained with a proper language model. The performance of our method with and without jointly training of WGAN-gp on sequential text generation was also evaluated. In order to evaluate the quality of generated text, we measured the perplexity of generated text and conducted human evaluation.

### 4.3.1 PERPLEXITY

The original English Gigaword and DBpedia datasets are already split into training set and testing set. We trained a general LSTM language model on the text of all training data, and for each class we trained a class LSTM language model initialized from general language model. The language models were then used to compute the perplexity of generated text. The perplexity of text on English Gigaword and DBpedia are shown in Table.4.

In both datasets, the perplexity of Topic GAN was almost as low as baseline method VAE+WGAN-gp, which suggested our method was able to generate equally smooth text. In addition, our method not only generated text with equal quality, it also generated text conditioned on discrete topic. As shown in Table.4, the perplexity of Topic GAN in class language model is lower than general language model. This implied that the generated text of topic GAN captured the information of each class. Jointly training also slightly improved the perplexity. As perplexity can not precisely reflect the quality of text, we conducted human evaluation to further discuss the performance of our model.

### 4.3.2 HUMAN EVALUATION

As there are no labeled categorical data on English Gigaword, we were not able to evaluate whether our model discovered meaningful latent topics. In addition, perplexity is not an accurate evaluation of the quality of the text. For the above two reasons, we conduct human evaluation. The human evaluation experiment was composed of two parts. The first part evaluated whether the text generated from the same latent topic can be recognized as the same class by human. In the second part evaluate the quality of the generated sentence.

Table 6: Topical words generated for discovered latent topics.

| Dataset | Topic ID | Topical Words |
|---|---|---|
| English Gigaword | 1 | midfielder, liverpool, munich, milan, defender, friendly, boss, debut, dallas, professional, portuguese, premiershipt |
| | 2 | #,###.##, benchmark, composite, hang, profit-taking, ##,###.##, stories, clients, investor, shenzhen |
| | 3 | killings, teenager, fbi, yemen, fatal, reportedly, jordanian, ring, transfer, cia, smuggling, atlanta |
| DBpedia | 1 | event, vocalist, songwriter, alpine, singing, pianist, bassist, piano, host, guitarist, rb, paintings |
| | 2 | streets, plaza, residence, mansion, cemetery, landmark, architectural, brick, chapel, revival, citys, surviving |
| | 3 | goalkeeper, midfielder, striker, defender, fc, defensive, footballer, matches, righthanded, loan, forward, soccer |
| Yahoo! Answers | 1 | crying, comfortable, glad, friendship, cheated, boyfriends, anyways, doi, shell, bother, jealous, talked |
| | 2 | crack, snow, boat, birds, sharing, aim, 98, fox, wood, syndrome, anti, monitor |
| | 3 | registered, purchased, certificate, fairly, entry, suit, applying, costs, lease, funds, tons, telephone |
| 20NewsGroups | 1 | solar, missions, spacecraft, shuttle, planetary, astronomy, orbit, satellites, materials, mars, flight, launch |
| | 2 | handguns, handgun, gun, violent, batf, firearms, criminals, firearm, sumgait, weapons, guns, mamma |
| | 3 | car, tires, engine, bike, ford, cars, ride, rear, seat, honda, miles, shop |

In the first part, the humans were demonstrated a set of example texts generated from the same latent topic and they were asked to categorize the example texts into a specific topic like sports,economy or violence. Then, following the example texts, there were several multiple-choice questions which had a text from same topic as correct choice and some texts from other topics as incorrect choice in each question. The humans were asked to choose the correct choice. As shown in Table 5(a), almost all human successfully selected the texts with the topic same as example class in each question. This result suggested that in the dataset without labeled categorical data, our method was able to discover meaningful latent topics and generate text from the discovered latent topic.

The second part measured the quality of generated text. The quality including the grammar and the rationality of the text. We asked human to select the better one among two sentences generated from different methods. In first two rows of Table 5(b), we compared the human preference of topic GAN with and without jointly training. The human preferred the text generated by topic GAN with jointly training to the text generated without jointly training. The last two rows of Table 5(b) showed that human slightly preferred topic GAN than baseline method(VAE+WGAN-gp).

## 5 CONCLUSION

This paper proposes TopicGAN, which can automatically capture topical information and generate the natural language with controled semantics in an unsupervised manner. The discrete representations show the superior performance on unsupervised classification, demonstrating the capacity of topic modeling in the proposed model. In addition, our model can generate texts with comparable performance with other approaches and additionally enable topical representation for better interpretation.

Table 7: Generated texts for different latent topics in Table 6.

| Dataset | Topic ID | Generated Text |
|---|---|---|
| English Gigaword | 1 | kobe bryant scored ## points to ## points as no. ## boston celtics beat the los angeles lakers #### to tie the nba champion |
| | 2 | tokyo stocks fell tuesday amid fears that consumer prices were cautious as inflation data drove in europe first ahead of a recession |
| | 3 | three soldiers were killed in a remote village in the north 's deadly suicide attack on tuesday |
| DBpedia | 1 | teeth on the rock is the only release by the japanese rock band ## on 25 march 201 |
| | 2 | the ## temple in portland oregon is located in ## n ## it is listed in the national register of historic places in 1998 |
| | 3 | ## ## born march 17 1986 is a lithuanian professional basketball player for the ## national basketball league |

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
