# OpenReview forum: "TopicGAN: Unsupervised Text Generation from Explainable Latent Topics"
_ICLR.cc/2019/Conference_

### Official Review · AnonReviewer2 · 2018-11-02
**not convincing**

**Rating:** 5
**Confidence:** 4

**Review:**

This paper proposes a new framework for topic modeling, which consists of two main steps: generating bag of words for topics and then using RNN to decode a sequence text.

Pros:
The author draws lessons from the infoGAN and designed a creative object function with reconstruction loss and categorical loss. As a result, this paper achieved impressive outcome for topic modeling tasks.

Comments:
1. High-level language is used to describe how to train two parts of the model, which is not technically clear. It would be better describe the algorithms in more details by listing steps for your algorithm in the section 3.3.

2. For text generation experiments, why didn’t you compare your model with any other related model such as SeqGAN or TextGAN? It is not so convincing to just use VAE+Wgan-gp as a baseline model.

3. For qualitative analysis part, you just listed some of your generated sentences for proving the fluency and relevance. Why didn’t you use some standard metrics for evaluating the quality of the text? I cannot judge the quality of your model through these randomly selected sentences.

4. As you mentioned in this paper “your model can be easily combined with any current text generation models”, have you done any experiments for demonstrating the original text generation model will get better performance after applying your framework?

Minor comments:
1. On page 2 and page 4, you mentioned “the third term in (2)”. According to my understanding, this should be equation 1 instead.

---

> ### Author Response · Authors · 2018-11-27
> **Thank you for your valuable review**
>
>
> (1)More details and writing:
> In the revised version of paper, we have provided more details and clearer explanations of our model in revised version Section 3.3. We have also rewritten many parts of the article to make the paper easier to understand.
>
> (2)Baseline:
> Because we use GAN to do the same fine tuning method  of our proposed TopicGAN and our baseline VAE+WGAN. Therefore, we consider it a proper baseline. Please notice that the baseline and other related works can only generate text conditioned on noise, while our generation task is more difficult that we conditioned not only on noise but also on discovered latent topics.
>
> (3)Evaluation metric:
> We used perplexity of generated text as our standard metrics. That is because we studied some previous generation papers(e.g.: https://arxiv.org/abs/1801.07736) and find some them using perplexity as their evaluation metric. To evaluate whether our topic model can discover meaningful topics, we also report topic coherence score in revised paper Table.3.
>
> (4)
> The reason that we said our method can be combined with any other text generation method using GAN is that we can use any other GAN to jointly train our whole text generator G, where G is composed of BOW generator $G_{bow}$ and sequence generator $G_{seq}$.
>
>
> Minor comments:
> You are right. It should be equation 1 instead. We have revised this mistake.

---

### Official Review · AnonReviewer3 · 2018-11-05
**This paper presents a topic model based on adversarial training.**

**Rating:** 4
**Confidence:** 4

**Review:**

This paper presents a topic model based on adversarial training. Specifically, the paper adopts the framework of InfoGAN to generates the bag-of-words of a document and the latent codes in InfoGAN correspond to the latent topics in topic modelling. In addition to the above framework, to make the model work better, several add-ons are also proposed, combining autoencoder, loss clipping, and a generative model to generate text sequences based on the bag-of-words.

My comments are as follows:

1. There are several issues of this paper on clarity:

(1) The first major one for me is that the authors did not give any details on how to interpret the latent code (i.e. the topics here) with the top words. In conventional topic models, usually a topic is a distribution of words, so that top words can be selected by their weights. But I did not see something similar in the proposed model.

(2) Another major one is why the word sequence generator is introduced in the proposed model. I did not see the contribution of this part to the whole model as a topic model, although the joint training shows the marginal performance gain on text generation.

(3) Some of the experiment settings are not provided, for example, the number of topics, the value of \alpha and \lambda in the proposed model, the hyperparameters of LDA, which are crucial for the results.

(4) Why is the size of the bag-of-words vocabulary set to be 3K whereas that of the word generation vocabulary set to be 15K?

Minor issues:

(5) In the related work of InfoGAN, there are a lot of cross-references to the following sections, before they are properly introduced.

(6) Typo of "Accurcay" in Table 4(a).

2. Using adversarial training for topic models seems to be an interesting idea. There is not much work in this line and this paper proposes a model that seems to be working. But it seems to be that the proposed model has several issues as follows:

(1) Each document seems to have only one topic, which can be an impractical setting for long documents.

(2) The proposed model ignores the word counts, which can be important for topic modelling.

(3) I did not see a major improvement of the proposed model over others, given that the only numerical result reported is classification accuracy and the state-of-the-art conventional topic models are not compared. This also leads to my concern about the experiments. I would expect more comparisons than classification accuracy, such as topic coherence and perplexity (for topic modelling) and with more advanced conventional models. From the low values of the accuracy on 20NG, I am wondering if LDA is working properly.

---

> ### Author Response · Authors · 2018-11-27
> **Thank you for your thorough review.**
>
>
> Writing:
> We have rewritten many parts of the article to make the paper easier to understand. In addition, some not convincing explanations mentioned in the review are also revised.
>
> (1)How to select topic words:
> Our topic classification model is a  V*K matrix M, where V is the word number and K is the number of latent topics.
> For each column of M is a topic distribution of words which is similar to conventional topic models such as LDA.
> The value of M[i][j] represents the importance of i-th word to j-th topic. Therefore, we were able to select the top few words with highest weight within each topic as topic words. We have included those details in the revised version (Section4.2).
>
> (2)Why word sequence generator is included in the paper:
> The goal of our work not only aims to train a high quality topic model, but also aims to generate high quality text using GAN by two steps generation. Using GAN for language generating is notorious for extremely difficult to train because it needs to (1) generate meaningful context with (2) correct grammar simultaneously. However, in our work, we try to separate this two core part of language generation and make the generation process easier.
>
> (3)Some detail:
> For all experiments(including baseline models), we set the topic number same as the class number. For example, in 20 News Groups, the class number is 20, and thus we set the topic number to 20. We use online LDA with different hyperparameters adjusted to get the better result on each dataset.
>
> (4)Why BOW vocabulary size is smaller:
> The size of the bag-of-words vocabulary is smaller because we hope during bag-of-words generation our model can focus on more important and general words. With smaller vocabulary size of bag-of-words, the result of unsupervised learning is better.
>
> (5)Cross-references:
> We have rewritten some methodology part and make it clearer.
>
> (6)Typo:
> We have revised this typo.
>
> Part2:
> (1)Assuming documents are generated from one single main topic:
> In our experiments, we conduct unsupervised document classification, in which the documents have only one single class. Therefore, for those unsupervised classification experiments, assuming each documents coming from a single main topic is a more appropriate assumption, which allows our model to learn more distinct topics. In addition, as the length of our training documents is short, it’s hard to break the short text into several topics, which is one of the possible reason that makes LDA works not well on short text.
>
> (2)Proposed model ignores the word counts:
> Although the model ignores the word counts, it still performs well in unsupervised document classification and topic coherence score.
>
> (3)Topic coherence score is reported:
> We have evaluated the topic coherence score and reported the score on revised paper Table 3. Our method outperformed baseline method on all datasets, which implies the effectiveness of our proposed topic model. We believe LDA worked properly as the topic coherence scores and unsupervised classification accuracy were in reasonable range.
> We think LDA is the most famous conventional topic model, could you list the state-of-the-art conventional topic model that should be compared?

---

### Official Review · AnonReviewer1 · 2018-11-07
**This paper proposes a generative adversarial approach to topic modeling. While the idea is fine, the paper has several limitations.**

**Rating:** 4
**Confidence:** 2

**Review:**

This paper proposes TopicGAN, a generative adversarial approach to topic modeling and text generation. The model basically combines two steps: first to generate words (bag-of-words) for a topic, then second to generate the sequence of the words.

While the idea is interesting, there are several important limitations. First, the paper is difficult to understand, and some of the explanations are not convincing. For example, in section 4.1.1, it says "... our method assumes that the documents are produced from a single topic ... Our assumption aligns well with human intuition that most documents are generated from a single main topic." This goes very much against the common assumption of a generative topic model, such as LDA, which the model compares against. I don't mean to argue either way, but if the paper presents a viewpoint which is quite different from the commonly accepted viewpoint (within the specific research field), then there needs to be a much deeper explanation, ideally with concrete evidence to support it. Another sentence from the same paragraph states that their "model outperforms LDA because LDA is a statistical model, while our generator is a deep generative model." This argument also seems flawed and without concrete evidence. There are other parts in the paper where the logic seems strange and without evidence, and they make it difficult to understand and accept the major claims of the paper.

Second, the model does not offer much novelty. It seems that the two-stage model simply puts the two pieces, a GAN-style generator and an LSTM sequence model together. Perhaps I am not understanding the model, but the model description was also not clear nor easy to understand with respect to its novelty.

Third, the evaluation is somewhat weak. There are two main evaluations tasks: text classification and text generation. For the first task, classification is not the main purpose of topic models, and while text classification _is_ used in many topic modeling papers, it is almost always accompanied by other evaluation metrics such as held-out perplexity and topic coherence. This is because the main purpose of topic modeling is to actually infer the topics (per-topic word distribution and per-document topic distribution) and model the corpus. Thus I feel it is not a fair evaluation to just compare the models using text classification tasks. The second evaluation task of text generation is not explained enough. For the human evaluation, who were the annotators, and how were they trained? How many people annotated each output, and what was the inter-rater agreement? How many sentences were evaluated, and how were they chosen? Without these details, it is difficult to judge whether this evaluation was valid.

Lastly, the results are mediocre. Besides the classification task, the others do not show significant improvements over the baseline models. Perplexity (table 3) shows similar results for DBPedia and worse results (than WGAN-gp) for Gigaword. Table 4 shows slightly better results for "Preference" for TopicGAN with joint training, but "Accuracy" is measured only for the proposed model and not the baseline model.

---

> ### Author Response · Authors · 2018-11-27
> **Thank you for your valuable review**
>
>
> (1)Writing:
> We have rewritten many parts of the article to make the paper easier to understand. In addition, some not convincing explanations mentioned in the review are also revised.
>
> (2)Assuming documents are generated from one single main topic:
> In our experiments, we conduct unsupervised document classification, in which the documents have only one single class. Therefore, for those unsupervised classification experiments, assuming each documents coming from a single main topic is a more appropriate assumption, which allows our model to learn more distinct topics. In addition, as the length of our training documents is short, it’s hard to break the short text into several topics, which is one of the possible reason that makes LDA works not well on short text.
>
> However, we acknowledge that for long documents, it's more appropriate to assume they come from the mixture of topics. In fact, it's feasible for our method to generate documents from several topics because info-GAN allows us to decide the distribution of the predicted code. We are conducting experiments on using several topics to generate longer documents and the current result seems better than generating from one single main topic.
>
> (3)Novelty:
> The novelty of our work is that (a) as far as we know, there is no previous work which tries to use GAN to achieve topic modeling, which is a worth exploring direction. (b) Some extra tricks for Info-GAN training (c)Two steps generation of text may also be a better and easier method for generating text.
>
> (4)Evaluation:
> We have evaluated the topic coherence score and reported the score on revised paper Table 3. Our method outperformed baseline method on all datasets, which implies the effectiveness of our proposed topic model.
> When conducting human evaluation to evaluate the quality of sentences, we asked 17 annotators to compare 13 sets of sentences generated by different methods.

---

### Meta-Review · Area_Chair1 · 2018-12-14
**technical details require clarification and experiments lack sufficient comparisons**

**Confidence:** 5
**Recommendation:** Reject

**Metareview:**

This paper proposes TopicGAN, a generative adversarial approach to topic modeling and text generation. TopicGAN operates in two steps: it first generates latent topics and produces bag-of-words corresponding to those latent topics. In the second step, the model generates text conditioning on those topic words.

Pros:
It combines the strength of topic models (interpretable topics that are learned unsupervised) with GAN for text generation.

Cons:
There are three major concerns raised by reviewers: (1) clarity, (2) relatively thin experimental results, and (3) novelty. Of these, the first two were the main concerns. In particular, R1 and R2 raised concerns about insufficient component-wise evaluation (e.g., text classification from topic models) and insufficient GAN-based baselines. Also, the topic model part of TopicGAN seems somewhat underdeveloped in that the model assumes a single topic per document, which is a relatively strong simplifying assumption compared to most other topic models (R1, R3). The technical novelty is not extremely strong in that the proposed model combines existing components together. But this alone would have not been a deal breaker if the empirical results were rigorous and strong.

Verdict:
Reject. Many technical details require clarification and experiments lack sufficient comparisons against prior art.